# BRIDGING AUTOREGRESSIVE AND MASKED MODELING FOR ENHANCED VISUAL REPRESENTATION LEARNING

## ABSTRACT

Autoregressive models have demonstrated superior performance in natural language processing due to their ability to handle large-scale training and generating ability. However, their potential in computer vision has not been fully explored due to some key challenges they still face. Currently, masked modeling methods such as MAE are dominant in this field. By analyzing autoregressive and masked modeling methods in a probabilistic way, we find that they can complement each other. Based on this, we propose a general formulation and modeling framework that combines the benefits of both, named **G**enerative **V**isual **P**retraining (GVP). Our unified probabilistic framework allows for different training strategies, including masked modeling and autoregressive modeling, to be realized simultaneously. Our framework can be adapted for various downstream tasks and outperform existing methods in several benchmarks, including linear probing, fine-tuning and transfer learning. This work provides a promising direction for future research in generative masked visual representation learning.

## 1 INTRODUCTION

Autoregressive models have gained popularity in natural language processing (NLP) due to their superior performance in large-scale training and generating abilities (Zeng et al., 2021; Brown et al., 2020; Zhang et al., 2021). However, their potential in the computer vision (CV) field is still largely unexplored. Although there have been attempts to treat images as sequences and utilize autoregressive modeling (Chen et al., 2020; Chang et al., 2022; Hua et al., 2023), masked modeling methods motivated by BERT (Devlin et al., 2018) like BEiT (Bao et al., 2022), MAE (He et al., 2022), and SimMIM (Xie et al., 2022) currently dominate and perform better in representation learning.

One of the key challenges that autoregressive models face in computer vision is their lack of natural order. While text has a clear and natural order, images do not, making it difficult to model them using autoregressive methods. This is due to the fact that images are two-dimensional, and the order in which pixels or patches are processed does not necessarily reflect the order in which information is conveyed in the image. As a result, certain tasks can be difficult to perform using autoregressive models, particularly those that require an understanding of the global structure of the image. In addition to the lack of natural order, images also have a higher degree of redundancy compared to text. This can make it easier to infer the target patch given much information (Xie et al., 2022). As a result, there is a gap in the difficulty of prediction between different parts in the autoregressive sequential order, which makes the learning process inefficient.

On the other hand, autoregressive models and masked modeling methods can complement each other, each possessing unique strengths. Autoregressive models can effectively capture intricate dependency relationships between tokens, while masked modeling methods treat predicted tokens independently regardless of sequential order. As proved in the field of NLP, the introduction of modeling intricate dependency by autoregressive modeling will assist masked modeling in learning more inter-sequence information and enhance its performance in generation tasks (Dong et al., 2019). On the contrary, masked modeling methods offer a flexible mask ratio, providing better control over task difficulty. Therefore, autoregressive models can still be effective in computer vision when combined with other methods to address the aforementioned issues. By combining these two methods, we can potentially improve the performance of various tasks.

To address these issues, we propose a general formulation and modeling framework that combines the benefits of both autoregressive and masked modeling methods. By using a unified probabilistic framework, we can simultaneously realize different training strategies including masked modeling and autoregressive modeling. This integration results in improved classification ability and flexible

generation. We conducted extensive experiments to validate the effectiveness of our framework on several benchmarks, including linear probing, fine-tuning and transfer learning. Our results demonstrate that our framework outperforms existing methods and provides a promising direction for future research in masked image modeling.

We summarize our contributions as follows:

- We bridge autoregressive and masked modeling methods through a probabilistic framework, named Generative Visual Pretraining (GVP) that combines the benefits of autoregressive and masked modeling methods.

- In GVP, we model various dependency relationship between tokens and realize different masking strategies using a unified Transformer module, which combines the benefits of autoregressive and masked modeling methods, leadings to improved classification ability and flexible generation.

- Our framework outperforms existing methods in several benchmarks, including linear probing, fine-tuning, and transfer learning.

## 2 RELATED WORK

### 2.1 MASKED MODELING

Masked Modeling is a task that learns by masking a portion of input signals and trying to predict these masked signals using the unmasked part. In natural language processing and computer vision fields, many approaches following this philosophy have achieved great performance. Masked language modeling (MLM) (Devlin et al., 2018; Liu et al., 2019; Joshi et al., 2020; Lan et al., 2019) like BERT are effective self-supervised learning approaches in the field of NLP. Given visible tokens in a sentence or a sentence pair, these approaches learn a bidirectional contextualized encoder for natural language understanding via denoising objectives. Masked Image Modeling (MIM) has located in a non-mainstream position for representation learning for a long time. Recently, BEiT (Bao et al., 2022), MAE (He et al., 2022), SimMIM (Xie et al., 2022) and MAGE (Li et al., 2022) recall this approach on Vision Transformer (Dosovitskiy et al., 2020) with various targets and demonstrated strong ability on representation learning. BEiT recovers discrete visual tokens from masked inputs in the pixel form and MAGE improves it by using discrete visual tokens as the input. On the contrary, MAE and SimMIM both recover raw pixels. Despite their embedding methods and target formulation, they share the same modeling paradigms with BERT, which maximize the likelihood of the predicted masked part given the unmasked part and assume each token in the masked part as mutually independent. However, this assumption will lead to poor reconstructive results (Bao et al., 2022; He et al., 2022).

### 2.2 AUTOREGRESSIVE MODELING

Autoregressive modeling is a widely used technique in NLP and CV fields, which predict the value of each token in the sequence based on the values of the tokens that came before it. Autoregressive modeling has been widely used in NLP to generate text, translate languages and perform sentiment analysis. One of the most prominent autoregressive models in NLP is Transformers Vaswani et al. (2017), which serves as the backbone of GPT series (Radford et al., 2018; 2019; Brown et al., 2020). For Transformers, an efficient way to train autoregressive models is through causal masks. It is used to ensure that the model generates text in a left-to-right fashion, meaning that it can only use information from earlier parts of the text to generate later parts. XLNet (Yang et al., 2019) further considers all possible permutations of the factorization order instead of using a fixed forward or backward factorization, where they propose two-stream self-attention to remove the ambiguity in target prediction. Since 2016, there have been works (Oord et al., 2016b;a) using autoregressive modeling to unconditionally generate images. However, among the ViT-based methods, autoregressive modeling is few used in representation learning. IGPT (Chen et al., 2020) directly use autoregressive modeling on pixels using transformer architecture, but it still mainly focuses on image generation. SAIM (Qi et al., 2022) and RandSAC (Hua et al., 2023) explore the autoregressive modeling in ViT but they do not explore how the group segmentation affects the modeling. Their prediction target and input form is also limited to raw pixel. There are still ways to explore the use of autoregressive modeling in CV's representation learning.

## 3 TOWARDS A UNIFIED PROBABILISTIC FORMULATION

Given an image $x$, the embedding layer in the ViT transfers the image into a sequence of tokens $\mathbf{x} = x_1 x_2 \ldots x_T$, where $T$ is the length of the sequence. Let $\mathcal{Z}_T$ be the set of all $T!$ permutations of the index sequence $[1, 2, \ldots, T]$ and let $\mathbf{z}$ denote one of the permutation. Denote $z_t$ the $t$-th element, $\mathbf{z}_{<t}$ the first $t-1$ elements of a permutation $\mathbf{z}$ and $\mathbf{z}_{>t}$ the last $T-t+1$ elements. Denote $\mathbf{x_y}$ a sub-sequence of $\mathbf{x}$ with the index sequence $\mathbf{y}$. We will first formalize the targets of masked modeling and autoregressive modeling in a probabilistic form and then unify them under one framework.

### 3.1 ANALYSIS IN THE PROBABILISTIC PERSPECTIVE

**For masked modeling,** we take MAE (He et al., 2022) as example. Review that MAE first masks $p \cdot T$ tokens of the sequence $\mathbf{x}_{\text{masked}}$ and get the visible part $\mathbf{x}_{\text{unmasked}}$, predicting the masked part through an encoder-decoder architecture $g \circ f$ using the visible part, where $p$ is the mask ratio, $f$ is the encoder and $g$ is the decoder. The loss function of one image is formulated by $\mathcal{L}_{\text{MAE}} = \mathbb{E}\|g \circ f(\mathbf{x}_{\text{unmasked}}) - \mathbf{x}_{\text{masked}}\|_2^2$, where the expectation is taken over on different choices of the masked portion and realized by randomly choosing one. From a probabilistic perspective, assume the output probability of reconstructed patches follow a conditional Gaussian distribution

$$p_\theta(\mathbf{x}_{\mathbf{z}_{\geq m}} | \mathbf{x}_{\mathbf{z}_{<m}}) = \mathcal{N}(\mu_{\mathbf{x}_{\mathbf{z}_{<m}}}, \mathbf{I}), \tag{1}$$

where $m$ is the length of the sequence $\mathbf{x}_{\text{unmasked}}$, $\mu_{\mathbf{x}_{\mathbf{z}_{<m}}} = g \circ f(\mathbf{x}_{\mathbf{z}_{<m}})$ is the mean vector and $\mathbf{I}$ is the identity covariance matrix. Then, we can easily see that the MAE objective is equivalent to the following negative log likelihood loss:

$$\mathcal{L}_{\text{MAE}} = -\mathbb{E}_{\mathbf{z}} \log p_\theta(x_{\mathbf{z}_{\geq m}} \mid \mathbf{x}_{\mathbf{z}_{<m}}) + C == -\mathbb{E}_{\mathbf{z}} \sum_{t=m}^{T} \log p_\theta(x_{z_t} \mid \mathbf{x}_{\mathbf{z}_{<m}}) + C. \tag{2}$$

The last equation is due to the fact that MAE implicitly assumes that the output patches are conditionally independent, i.e., $p_\theta(x_{z_i}|\mathbf{x}_{\mathbf{z}_{<m}}) \perp p_\theta(x_{z_j}|\mathbf{x}_{\mathbf{z}_{<m}}), \forall i, j \geq m$. In other words, the generation process from $\mathbf{x}_{\mathbf{z}_{<m}}$ to $\mathbf{x}_{\mathbf{z}_{\geq m}}$ is fully non-autoregressive (Luong & Manning, 2015). This assumption could be very restrictive, particularly when adopting a very large mask ratio – which helps explain the poor image reconstruction quality shown in MAE (He et al., 2022; Dong et al., 2022).

**For autoregressive modeling,** we take iGPT (Chen et al., 2020) as example. IGPT considers each image pixel as a token and fix the index sequence $\mathbf{z} = [1, 2, \ldots, T]$, i.e., from left to right and from up to down. During the training stage, iGPT predicts each token using all tokens before itself in the sequential order with a transformer network $f$. Since all tokens are pixels, the prediction task is a classification task with the cross entropy loss. The loss function of one image is formulated by

$$\mathcal{L}_{\text{iGPT}} = \sum_{t=1}^{T} \text{CE}(f(\mathbf{x}_{\mathbf{z}_{<t}}), x_{z_t})). \tag{3}$$

Here, denoting the softmax output $\mu_t = f(\mathbf{x}_{\mathbf{z}_{<t}})$ as the probability distribution of a categorical distribution $p_\theta(x_t|\mathbf{x}_{\mathbf{z}_{<t}}) = \text{Cat}(\mu)$, we can show that this objective also corresponding to the following log likelihood with autoregressive factorization:

$$\mathcal{L}_{\text{iGPT}} = \log p_\theta(\mathbf{x}) = \sum_{t=1}^{T} \log p_\theta(x_{z_t} \mid \mathbf{x}_{\mathbf{z}_{<t}}). \tag{4}$$

**Comparison.** Comparing the two paradigms for MIM, we notice that each paradigm has its own advantages and shortcomings. For masked modeling, the index sequence $\mathbf{z}$ is not fixed, which enables the model to explore broader relation between tokens. On the contrary, the index sequence $\mathbf{z}$ for autoregressive modeling is fixed, **resulting in the lack of flexibility of the modeling of the data distribution**, which could be harmful in representation learning, where it is crucial to incorporate context from both directions. Besides, unlike the situation in NLP fields, there is no natural sequence order for images. Therefore, the fixed order adopted by iGPT may not be optimum. Second, the task complexity for all predicted tokens are adjustable by altering different mask ratio, which adjust the $m$ in Equation (2). This allows the model to vary the task difficulty between different image samples, which can help the model achieve better performance on many downstream tasks (Li et al., 2022; Chang et al., 2022). However, the fix index sequence will fix the task difficulty in autoregressive task among different image samples, which will also lead to the inflexibility of the modeling of

data distribution. There is also a problem that the prediction tasks for the tokens in the end of the sequence are too easy, leading to the decrease of training efficiency. Third, autoregressive modeling build vigorous dependency between tokens, providing a natural way to use the product rule for factorizing the joint probability of the predicted tokens, eliminating the independence assumption made in masked modeling.

From the above observation, it can be seen that masked modeling and autoregressive modeling are two complementary approaches for masked image modeling. In the next section, we explore a general probabilistic formulation for a principled combination of the two generative learning paradigms.

## 3.2 A General Framework for Generative Masked Visual Representation Learning

In this section, we establish a unified probabilistic framework that unifies masked and autoregressive modelling. It could enable a general and flexible way to learn visual representation learning in a generative approach.

We start by a generative modeling of the joint data distribution. Review that the objective of masked modeling is to maximize the log-likelihood of the last part of the sequence conditioned on the front part, i.e. $\log p_\theta(\mathbf{x}_{\mathbf{z} \geq m} \mid \mathbf{x}_{\mathbf{z} < m})$ , while the objective of autoregressive modeling is to maximize the log-likelihood of the data sample $\log p_\theta(\mathbf{x})$. Notice that $\log p_\theta(\mathbf{x})$ is just a specific example of $\log p_\theta(\mathbf{x}_{\mathbf{z} \geq m} \mid \mathbf{x}_{\mathbf{z} < m})$ where $m = 1$. Additionally, there can be some extra condition information $\mathbf{y}_{\text{extra}}$ provided in the task, such as the source image in the style transforming task (Hu et al., 2022) and the class label in the conditional generation task (Oord et al., 2016b), which can be both integrated in the condition part of the probability function. Therefore, we select $\log p_\theta(\mathbf{x}_{\mathbf{z} \geq m} \mid \mathbf{x}_{\mathbf{z} < m}, \mathbf{y}_{\text{extra}})$ as our basic formulation.

Further, in order to include both masked modeling and autoregressive modeling, the new framework should satisfy the following requirements: (1) The new framework is able to model on any sequence order. (2) The new framework is able to flexibly adjust task complexity. (3) The new framework is able to model vigorous dependency between tokens. We will show how to build this framework starting from the basic probabilistic goal.

First, to be flexible enough to model any sequence order, the $\mathbf{z}$ should not be fixed but chosen randomly in a certain way. Hence, the objective becomes $\mathbb{E}_{\mathbf{z}} \log p_\theta(\mathbf{x}_{\mathbf{z} \geq m} \mid \mathbf{x}_{\mathbf{z} < m}, \mathbf{y}_{\text{extra}})$, where $\mathbf{z}$ obeys a certain distribution on the permutation space $\mathcal{Z}_T$. To satisfy the third requirement, we should adopt the method of autoregressive modeling on decomposing $\log p_\theta(\mathbf{x}_{\mathbf{z} \geq m} \mid \mathbf{x}_{\mathbf{z} < m}, \mathbf{y}_{\text{extra}})$, that is, to use the chain rule to factorize the probability. However, as mentioned before, simply adopting the autoregressive modeling will lead to problems on task complexity. Therefore, instead of completely decomposing the joint probability into tokens, we can extend the standard token-level chain rule to the group-level chain rule. To be specific, we can segment the index sequence $\mathbf{z}$ into $K + 1$ consecutive groups $G_0, G_1, G_2, \ldots, G_K = (z_1 \ldots z_{n_0}), (z_{n_0+1} \ldots z_{n_1}), (z_{n_1+1} \ldots z_{n_2}), \ldots, (z_{n_{K-1}+1} \ldots z_{n_K})$, where $m - 1 = n_0 < n_1 < n_2 < \cdots < n_K = T$. The conditional log-likelihood of data sample can be written as

$$
\begin{aligned}
\log p_\theta(\mathbf{x}_{\mathbf{z} \geq m} \mid \mathbf{x}_{\mathbf{z} < m}, \mathbf{y}_{\text{extra}}) &= \sum_{k=1}^{K} \log p_\theta(\mathbf{x}_{G_k} \mid \mathbf{x}_{G_0}, \mathbf{x}_{G_1}, \ldots, \mathbf{x}_{G_{k-1}}, \mathbf{y}_{\text{extra}}) \\
&= \sum_{k=1}^{K} \left( \sum_{t=n_{k-1}+1}^{n_k} \log p_\theta(x_t \mid \mathbf{x}_{\mathbf{z}_{<n_{k-1}}}, \mathbf{y}_{\text{extra}}) \right).
\end{aligned}
\tag{5}
$$

The formulation in Equation (5) is similar to that of autoregressive modeling, but with tokens predicted groups by groups. Consecutive elements in the same group are considered as conditionally independent like in masked modeling. The selection of $K$ and $n_k, 0 \leq k \leq K - 1$ can be flexibly adjusted to get different probabilistic formulation. Larger $n_i$ will lead to richer condition and less tokens to predict, resulting in easier task. Therefore, by adjusting $n_k, 1 \leq k \leq K - 1$, the task complexity can be correspondingly adjusted, which enable the formulation satisfy the second requirement. Specifically, setting $n_{K-1}$ to a smaller number will avoid the problem appearing in autoregressive modeling that the prediction tasks for the last few tokens in the sequence are too easy. Here we propose three possible settings of the group segmentation:

- **Fixed Length.** Let $n_i = [(i + 1)T/(K + 1)], i = 0, 1, 2, \ldots, K$, where each group has around the same $[T/(K + 1)]$ elements. The group segmentation is fixed and uniform in this setting, the same as autoregressive modeling;

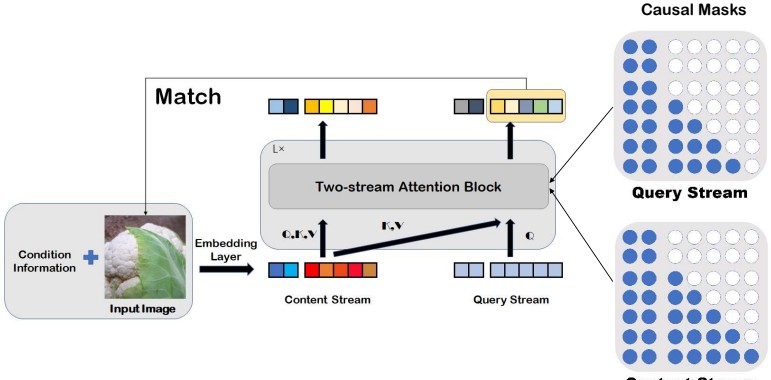

Figure 1: The overall pipeline of GVP: The input image and conditional information are embedded through an embedding layer to get the initial feature in the content stream. The initial feature in the query stream is a learnable vector. The features in the two stream are passed through two-stream attention blocks with causal masks. The output from the last layer of the query stream is used to match the input image. In the causal masks, blue for 0 and white for $-\infty$. More visualizations of the attention masks are shown in the appendix.

- **Random Length.** Let $n_0, n_1, \ldots, n_{K-1}$ uniformly distributed in the range $[1, T-1]$. This setting adds some randomness compared to the first setting, which can explore more dependency relationship between tokens.

- **Mixed Length.** Let $n_{K-1} = p \cdot T$ with $p \sim \mathcal{N}(0.5, 0.1)$ and let $n_0, n_1, \ldots, n_{K-2}$ uniformly distributed in the range $[1, n_{K-1} - 1]$ (more details are in the appendix). In this way, the tokens in the last group $G_K$ can only be depended on $p \cdot T$ tokens, which has a low probability to be larger than $0.8T$, avoiding the problem of too easy prediction tasks. Meanwhile, we can build vigorous dependency between the first $n_{K-1}$ tokens by choosing larger $K$. We set this as the default setting during training and take $K$ as 20.

In all, we have built a general probabilistic formulation that combines the merits of two modeling methods, whose objective can be written as

$$\mathcal{L}_{\text{general}} = \mathbb{E}_{\mathbf{z}, K, n_1, \ldots, n_K} \sum_{k=1}^{K} \left( \sum_{t=n_{k-1}+1}^{n_k} \log p_\theta \left( x_t \mid \mathbf{x}_{\mathbf{z}_{<n_{k-1}}}, \mathbf{y}_{\text{extra}} \right) \right), \tag{6}$$

where the expectation taken here means $\mathbf{z}, K, n_1, n_2, \ldots, n_K$ obey a certain distribution, that is, there can be a selecting strategy for $\mathbf{z}, K, n_1, n_2, \ldots, n_K$. We name this framework **G**enerative **V**isual **P**retraining (GVP). In the next section, we will introduce the way to implement this formulation in real training process.

## 4 IMPLEMENTATION VIA UNIVERSAL CAUSAL MASKS

In this section, we will discuss how to implement the general formulation in a single unified model. We will first propose out overall design and then illustrate our key design on the universal causal masks in detail.

### 4.1 OVERALL DESIGN

As shown in the overall pipeline Figure 1, the whole pipeline of our proposed GVP is composed by the following steps.

**Input Embedding.** First, the image $x \in \mathbb{R}^{H \times W \times C}$ is converted to an embedding sequence along with all the extra conditional information $\mathbf{y}_{\text{extra}}$ using an embedding layer, i.e. $e(\mathbf{y}_{\text{extra}}, x) = [y_1; \ldots; y_S; x_1; \ldots; x_T]$, where the extra conditional information is transformed into $[y_1; \ldots; y_S] \in \mathbb{R}^{S \times D}$ and the image is transformed into $[x_1; \ldots; x_T] \in \mathbb{R}^{T \times D}$. For the embedding layer, most existing methods simply flatten the image into a sequence of image patches and use a linear layer to get

the feature (Dosovitskiy et al., 2020; He et al., 2022; Xie et al., 2022; Bao et al., 2022). There are also methods using discrete encoder like VQGAN (Esser et al., 2021) to tokenize the image (Li et al., 2022). We will consider both in our implementation. After adding positional embedding $\mathbf{E}_{pos} \in \mathbb{R}^{(S+T) \times D}$, we get the feature before the first layer of the content stream $\mathbf{H}^{(0)} = [y_1; \ldots; y_S; x_1; \ldots; x_T] + \mathbf{E}_{pos}$. The feature before the first layer of the query stream is initialized with a trainable vector $w$, i.e. $\mathbf{G}^{(0)} = [w; w; \ldots; w] \in \mathbb{R}^{(S+T) \times D}$.

**Backbone Network.** In our objective of Equation 6, the general formulation includes using various index sequences $\mathbf{z}$ and vigorous dependency between tokens, which both describe the attending relationship between tokens, i.e., which token can be attended to another given token. Therefore, it is natural to model the probability $p_\theta$ using a ViT architecture (Dosovitskiy et al., 2020) with causal masks. However, we are not capable of modeling with arbitrary index sequence in an autoregressive way if we use a single stream in ViT with causal masks. This is because of two requirements that are contradictory in a standard ViT architecture, which have been discussed in XLNet (Yang et al., 2019): (1) Suppose the output of the network in position $z_t$ is parameterized by $g_\theta(x_{\mathbf{z}<t}, z_t)$. In order to predict the token $x_{z_t}$, the network output in the $z_t$ position in one layer $g_\theta(x_{\mathbf{z}<t}, z_t)$ should only use the position $z_t$ and not the content $x_{z_t}$, (2) to predict the other tokens $x_{z_j}$ with $j > t$, $g_\theta(x_{\mathbf{z}<t}, z_t)$ should also encode the content $x_{z_t}$ to provide full contextual information. Therefore, it is not enough with only one stream. Since the two-stream attention proposed by XLNet facing the same problem works well, we build upon the two-stream attention mechanism proposed by XLNet in Section 4.2 and extend it by excavating important usage of the causal masks in Section 4.3.

**Loss Function.** The representation $\mathbf{G}^L$ in the query stream after the last layer is then trained to match the target. We have two formulations on how to select the target and calculate the loss. The first formulation is to convert the output into raw pixels using a linear layer and an MSE loss is used to measure the reconstruction error (He et al., 2022; Xie et al., 2022). The second formulation is to obtain a discrete class of the target by some discrete tokenizer and use a cross entropy loss between this ground-truth one-hot token and the output of the ViT (Bao et al., 2022; Li et al., 2022), where we will specifically use VQGAN (Esser et al., 2021) as the tokenizer in our implementation.

## 4.2 TWO-STREAM ATTENTION FOR GENERATIVE IMAGE MODELING

In this section, we will build upon the two-stream attention and propose our general design for realizing the generative image modeling. In the two-stream attention Transformer, the content stream $h_\theta$ encodes the contextual information, and the query stream $g_\theta$ predicts the targets with the help of the content stream. The two stream share the same weights of the attention block but differ in the causal mask. Intuitively, the causal masks in the two streams enable us to easily formulate various dependency relationship between tokens. The main difference for the two causal masks is that the content stream should ensure $x_{z_t}$ can be attended to itself while the query stream is the contrary. The two-stream attention allows us to arbitrarily change the sequential order without confronting the inconsistency as mentioned above. For each self-attention layer $l = 1, \ldots, L$, the two streams of representations are schematically updated with a shared set of parameters. In the $l$-th layer, the outputs of a self-attention head $\mathbf{A}_h^{(l)}$ and $\mathbf{A}_g^{(l)}$ in the two-stream are computed in the form of:

$$\mathbf{Q}_h = \mathbf{H}^{(l-1)} \mathbf{W}_Q^l, \ \mathbf{K}_h = \mathbf{H}^{(l-1)} \mathbf{W}_K^l, \ \mathbf{V}_h = \mathbf{H}^{(l-1)} \mathbf{W}_V^l$$
$$\mathbf{Q}_g = \mathbf{G}^{(l-1)} \mathbf{W}_Q^l, \ \mathbf{K}_g = \mathbf{H}^{(l-1)} \mathbf{W}_K^l, \ \mathbf{V}_g = \mathbf{H}^{(l-1)} \mathbf{W}_V^l \tag{7}$$
$$\mathbf{A}_h^{(l)} = \mathrm{softmax}(\frac{\mathbf{Q}_h \mathbf{K}_h^\top}{\sqrt{d_k}} + \mathbf{M}_h) \mathbf{V}_h, \mathbf{A}_g^{(l)} = \mathrm{softmax}(\frac{\mathbf{Q}_g \mathbf{K}_g^\top}{\sqrt{d_k}} + \mathbf{M}_g) \mathbf{V}_g,$$

where $\mathbf{H}^{(l-1)}$ and $\mathbf{G}^{(l-1)}$, the representations before the $(l-1)$-th layer of the content stream and the query stream, are linearly projected to queries, keys and values using shared trainable parameter matrices $\mathbf{W}_Q^l, \mathbf{W}_K^l, \mathbf{W}_V^l$ respectively, and $d_k$ is the dimension of the representations. $\mathbf{M}_h, \mathbf{M}_g \in \mathbb{R}^{(S+T) \times (S+T)}$ are the causal masks in the content stream and the query stream, respectively, which are the key factors in our framework. The value of each element $(\mathbf{M})_{ij}$ in either causal mask can only be 0 or $-\infty$, where $(\mathbf{M})_{ij} = 0$ means the $j-$th token **can** be attended to the $i-$th token and $(\mathbf{M})_{ij} = -\infty$ means the $j-$th token **cannot** be attended to the $i-$th token. This indicates that the causal mask has a one-to-one correspondence with the dependency relationship between tokens. Therefore, **we only need to alter the causal masks** $\mathrm{M}_h$ **and** $\mathrm{M}_g$ **to adapt to different choices of** $\mathbf{z}, K, n_0, \ldots, n_{K-1}$**, which eliminate the need of using multiple architectures.** We will show how to construct the corresponding attention masks in the next section.

### 4.3 Universal Causal Masks

In this section, we will show how to design causal masks $\mathbf{M}_h$ and $\mathbf{M}_g$ to suit various target formulations inside the expectation in Equation (6). Review that the value of each element $(\mathbf{M})_{ij}$ in either causal mask can only be $0$ or $-\infty$, where $(\mathbf{M})_{ij} = 0$ means the $j-$th token **can** be attended to the $i-$th token and $(\mathbf{M})_{ij} = -\infty$ means the $j-$th token **cannot** be attended to the $i-$th token. This implicates that by finding out the dependency between each token we can determine the causal mask.

We will start from the simplest formulation

$$-\log p_\theta(\mathbf{x}_{\mathbf{z} \geq 1} \mid \mathbf{y}_{\text{extra}}) = -\sum_{t=1}^{T} \log p_\theta(x_t \mid \mathbf{x}_{\mathbf{z} < t}, \mathbf{y}_{\text{extra}}). \tag{8}$$

where the index sequence is selected to be $[1, 2, \ldots, T]$ and each group has only one element. In this situation, a certain condition information token can be attended to all $x_t, 1 \leq t \leq T$ and $y_s, 1 \leq s \leq S$, but does not depend on any $x_t$. Therefore, for both $(\mathbf{M}_h)_{ij}$ and $(\mathbf{M}_g)_{ij}$, the value equals $0$ if $j \leq S$ and equals $-\infty$ if $j > S$ and $i \leq S$. Besides, every $x_i$ depends on $x_j$ with $i > j$. In addition, in order to build contextual information, every $x_i$ should depends on itself in the content stream. In conclusion, the causal masks for the two streams should be

$$(\mathbf{M}_h)_{ij} = \begin{cases} 0 & j \leq S \\ -\infty & j > S \geq i \\ 0 & i \geq j > S \\ -\infty & j > i > S \end{cases}, \quad (\mathbf{M}_g)_{ij} = \begin{cases} 0 & j \leq S \\ -\infty & j > S \geq i \\ 0 & i > j > S \\ -\infty & j \geq i > S \end{cases} \tag{9}$$

which is also visualized in Figure 1. If we consider putting a part of $\mathbf{x}$ into the condition and get the formulation

$$-\log p_\theta(\mathbf{x}_{\mathbf{z} \geq m} \mid \mathbf{x}_{\mathbf{z} < m}, \mathbf{y}_{\text{extra}}) = -\sum_{t=m}^{T} \log p_\theta(x_t \mid \mathbf{x}_{\mathbf{z} < t}, \mathbf{y}_{\text{extra}}), \tag{10}$$

the new causal masks can be attained by modifying the $S$ to $S + m - 1$ since the first $m$ tokens of $\mathbf{x}$ are integrated into the condition. Then we consider the general situation where tokens are segmented into groups as in Equation (5). Let $f(t)$ denote the index of the group to which $x_t$ belongs. In this situation, $x_i$ will be dependent on $x_j$ if $f(i) > f(j)$. Besides, tokens in the same group is dependent on each other in the content stream to provide contextual information. Therefore, note that $x_i$ is the $(S + i)$-th token in the whole sequence, by additionally defining $f(t)$ to be $0$ with non-positive $t$, the causal masks for the two streams should be

$$(\mathbf{M}_h)_{ij} = \begin{cases} 0 & j \leq S + m - 1 \\ -\infty & j > S + m - 1 \geq i \\ 0 & f(i - S) \geq f(j - S) > 0 \\ -\infty & f(j - S) > f(i - S) > 0 \end{cases}, \quad (\mathbf{M}_g)_{ij} = \begin{cases} 0 & j \leq S + m - 1 \\ -\infty & j > S + m - 1 \geq i \\ 0 & f(i - S) > f(j - S) > 0 \\ -\infty & f(j - S) \geq f(i - S) > 0 \end{cases} \tag{11}$$

At last, we consider varying the index sequence $\mathbf{z}$. To fit an given order $\mathbf{z} = z_1 z_2 \ldots z_T$, we only need to swap the columns and the rows of the original causal masks with order $[1, 2, \ldots, T]$, which we assume to be $\mathbf{M}_h$ and $\mathbf{M}_g$. Let $z'(i)$ be the reversed sequence map of $z$, i.e., $z'(i) = i$ if $i \leq S$ and $z'(z_i + S) = i + S$. Then we can obtain the causal masks $\mathbf{M}_h^{\mathbf{z}}$ and $\mathbf{M}_g^{\mathbf{z}}$ with index sequence $\mathbf{z}$ by:

$$(\mathbf{M}_h^{\mathbf{z}})_{ij} = (\mathbf{M}_h)_{z'(i)z'(j)}, \quad (\mathbf{M}_g^{\mathbf{z}})_{ij} = (\mathbf{M}_g)_{z'(i)z'(j)} \tag{12}$$

By now, we have constructed the causal masks for all possible dependency relations between tokens in our framework.

## 5 Experiments

We conduct self-supervised pretraining on the ImageNet-1K (Deng et al., 2009) training set. Then we conduct experiments under two protocols: first is linear probing, where we train a classification head on top of the learned representations while keeping the backbone frozen; second is fine-tuning, where we fine-tune the whole parameters for the classification task. We also include results on transfer learning to better evaluate the quality of the representations. More results and ablation studies can be found in the Appendix.

To fully validate our proposed framework, we conduct experiments with three variants of GVP under three settings of embedding layer and target mentioned in Section 4.1: (1) GVP-p2p (pixel-to-pixel):

Table 1: Top-1 accuracy of linear probing and fine-tuning on ImageNet-1k. Auto. is short for autoregressive modeling. MIM is short for masked image modeling. LP Acc. is short for linear probing accuracy. FT Acc. is short for fine-tuning accuracy. The number of parameters for MAGE and GVP-d2d includes VQ-GAN tokenizer and ViT encoder.

| Methods | Paradigm | Model | #params | Epochs | LP Acc.(%) | FT Acc.(%) |
|---|---|---|---|---|---|---|
| Scratch on Pixels | - | ViT-B | 86M | - | 27.3 | 77.1 |
| Scratch on Tokens | - | ViT-B | 24M+86M | - | 37.1 | 75.3 |
| MaskGIT(Chang et al., 2022) | MIM | BERT | 227M | 300 | 63.1 | - |
| iGPT(Chen et al., 2020) | Auto. | iGPT-L | 1362M | 100 | 60.3 | 66.3 |
| BEiT(dVAE)(Bao et al., 2022) | MIM (p2d) | ViT-B | 86M | 200 | 52.1 | 82.4 |
| BEiT(VQGAN) | MIM (p2d) | ViT-B | 86M | 200 | 54.3 | 82.6 |
| MAE(He et al., 2022) | MIM (p2p) | ViT-B | 86M | 200 | 55.3 | 82.9 |
| SimMIM(Xie et al., 2022) | MIM (p2p) | ViT-B | 86M | 200 | 52.5 | 82.8 |
| MAGE(Li et al., 2022) | MIM (d2d) | ViT-B | 24M+86M | 200 | 67.2 | 81.5 |
| RandSAC(Hua et al., 2023) | Auto. (p2p) | ViT-B | 86M | 200 | 58.2 | 83.0 |
| GVP-p2p | Auto. & MIM | ViT-B | 86M | 200 | 61.4 | 83.2 |
| GVP-p2d | Auto. & MIM | ViT-B | 86M | 200 | 58.3 | **83.4** |
| GVP-d2d | Auto. & MIM | ViT-B | 24M+86M | 200 | **69.8** | 81.7 |

linear projection as the embedding layer and pixel reconstruction as the target; (2) GVP-p2d (pixel-to-discrete): linear projection as the embedding layer and discrete classification as the target; (3) GVP-d2d (discrete-to-discrete): VQGAN tokenizer as the embedding layer and discrete classification as the target. Since there is no extra information in this scene, we just replace the $\mathbf{y}_{extra}$ with a class token. The index sequence $\mathbf{z}$ is uniformly selected in all permutations. For the embedding layer of linear projection, we set the input image resolution as 224 and the token sequence length is $14 \times 14$. For the embedding layer of VQGAN tokenizer, we set it as $256 \times 256$. After passing through the embedding layer, the token sequence length is $16 \times 16$. Following MAE (He et al., 2022), we use random resize crop (with rate of 0.2 to 1) and random flipping as our default augmentations. The default architecture is ViT-B and we use AdamW (Loshchilov & Hutter, 2019) to train the model for 200 epochs with batch size of 4096. We use a cosine learning rate schedule (Loshchilov & Hutter, 2017) with 10-epoch warm-up. The base learning rate is $1.5 \times 10^{-4}$ and scaled by batch size$/256$. For linear probing, we keep the causal mask to maintain the same feature distribution with pretraining, and we drop the causal mask when we fine-tune the model for a larger receptive field. Our proposed flexible network design integrates the modeling paradigm from both mask modeling and autoregressive tasks, which is proved to be beneficial to various downstream tasks in the field of NLP (Yang et al., 2019). The flexible attention masks enable the pretrained model suitable for both classification task (Section 5.1) and generation task (Appendix F).

## 5.1 IMAGE CLASSIFICATION

**Linear Probing.** Linear probing is a common evaluation protocol for self-supervised learning. We compare our results with various masked modeling methods and autoregressive methods. For methods using ViT, we mark their setting of embedding layer and target in order to fairly compare. The original BEiT uses dVAE (Vahdat et al., 2018) to get the target and we conduct BEiT using VQGAN to match the setting. As shown in Table 1, GVP-d2d outperforms MAGE by 2.6% for ImageNet-1K linear probe top-1 accuracy, which achieve state-of-the-art results among all mentioned methods, which proves the effectiveness of the general framework. Besides, GVP-p2p surpasses MAE and simMIM by at least 6% and GVP-p2d surpasses BEiT(VQGAN) by 4.0%, where the methods compared in the same group share the same embedding layer and target. This further indicates the superiority of our modeling on the dependency of the tokens.

**Fine-tuning.** Table 1 also hows the fine-tuning performance of GVP and other methods. Our GVP-p2p outperforms the autoregressive method RandSAC by 0.4% and the masked modeling method MAE by 0.5%. GVP-p2d also has a improvement of 0.8% over BEiT(VQGAN). These results show the strong representation learning ability of our modeling.

**Transfer Learning.** Following the protocol in the work of Ericsson et al. (2021) and Zhao et al. (2023), we evaluate the transfer learning performance pretrained on ImageNet-1K on 9 downstream datasets, which are FGVC Aircraft (Maji et al., 2013), Caltech-101 (Fei-Fei et al., 2004), Stanford Cars (Krause et al., 2013), CIFAR10(Krizhevsky, 2009), CIFAR100(Krizhevsky, 2009), DTD(Cimpoi et al., 2014), Oxford 102 Flowers (Nilsback & Zisserman, 2008), Food-101 (Bossard et al., 2014) and Oxford-IIIT Pets (Parkhi et al., 2012). For linear evaluation, multinomial logistic regression is fit on the extracted features. Results are shown in Table 2. We see that GVP-p2p outperforms MAE by 4.73% in average and GVP-p2d outperforms BEiT(VQGAN) by 2.77% in average. This results again

Table 2: Transfer learning performance of ViT-B pretrained on ImageNet-1K using different methods. Linear probing accuracy (%) is adopted. The **Avg** takes the average of the performance of nine datasets.

|  | Aircraft | Caltech101 | Cars | CIFAR10 | CIFAR100 | DTD | Flowers | Food | Pets | **Avg** |
|---|---|---|---|---|---|---|---|---|---|---|
| MAE | 81.09 | 82.54 | 82.61 | 90.12 | 66.82 | 64.22 | 96.17 | 76.54 | 77.24 | 79.71 |
| GVP-p2p | 83.56 | 86.19 | 87.28 | 96.12 | 73.41 | 71.44 | 95.89 | 76.78 | 89.30 | 84.44 |
| BEiT(VQGAN) | 78.24 | 81.13 | 76.24 | 88.14 | 67.13 | 66.25 | 94.44 | 74.69 | 82.32 | 78.73 |
| GVP-p2d | 80.97 | 83.93 | 77.13 | 92.98 | 70.44 | 72.87 | 95.77 | 73.32 | 86.12 | 81.50 |

Table 3: Linear probing accuracy and fine-tuning accuracy of GVP-p2p with fixed index sequence $\mathbf{z} = [1, 2, \ldots, T]$ and random index sequence.

| Index Sequence | Epochs | LP Acc.(%) | FT Acc.(%) |
|---|---|---|---|
| Fixed | 200 | 58.6 | 82.7 |
| Random | 200 | 61.4 | 83.2 |

indicate that under the same setting of embedding layer and target, our modeling is superior to the conventional masked modeling.

## 5.2 ABLATION STUDY

In this section, we analyze the key component of GVP framework: the index sequence and the dependency relationship between tokens. All experiments are conducted on ViT-B using the setting of linear projection and pixel reconstruction. We also include ablation study exploring the key factor in our proposed methods in Appendix H.

**Patch Ordering.** We first study the effect of unfixed index sequence. As shown in Table 3, GVP with random index sequence performs better than that with fixed index sequence in both protocols under 200 epochs pretraining, indicating that modeling with random index sequence will help the model better capture bidirectional contextual information.

**Analysis on Auto-regressivity.** We find that the quality of the learned representation can be greatly affected by the setting of group segmentation. We conduct experiments using the three settings mentioned in Section 3.2 and explore the best segmentation amount $K$. As shown in Table 4, modeling with mixed length will bring the best performance. Group segmentation with fixed length will lead to low accuracy, which is because fixed length modeling is lack of flexibility, disabling the model building more dependency relationship between tokens.

Table 4: Linear probing accuracy and fine-tuning accuracy of GVP-p2p with different group segmentation setting.

| Segmentation setting | $K$ | LP Acc.(%) | FT Acc.(%) |
|---|---|---|---|
| Fixed Length | 1 | 51.2 | 82.1 |
|  | 5 | 53.2 | 82.6 |
|  | 20 | 54.4 | 82.5 |
|  | 40 | 54.2 | 82.5 |
|  | 196 | 52.2 | 82.3 |
| Random Length | 5 | 57.8 | 82.7 |
|  | 20 | 55.6 | 82.4 |
|  | 40 | 56.4 | 82.5 |
| Mixed Length | 5 | 58.4 | 82.7 |
|  | 20 | **61.4** | **83.2** |
|  | 40 | 60.8 | 83.1 |

## 6 CONCLUSION

In this paper, we proposed a generative visual pretraining framework named GVP. In particular, we formally analyzed autoregressive and masked modeling methods in a probabilistic way. We proposed a general formulation and modeling framework combining the benefits of both, which focused on the dependency relationship between tokens. Specifically, we designed universal causal masks based on the two-stream attention to implement all the formulation using only one architecture. Empirically, we compared our proposed framework in the accuracy of linear probing, finetuning and transfer learning. We found that our modeling on the dependency relationships between tokens improved representation learning ability. We hope that our general framework can provide new insights for the community and inspire future works.

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

## A  EXAMPLES AND EXPLANATION ON CAUSAL MASKS

In this section, we will provide more details and examples on the design of causal masks under different formulations. We assume that $\mathbf{y}_{\text{extra}} = [y_1; y_2]$ and $\mathbf{x} = [x_1; x_2; x_3; x_4; x_5]$. Therefore, we have $S = 2$ and $T = 5$. We will provide the full causal masks with this example **when reviewing the reasoning process in Section 4.3.** For the simplest formulation

$$-\log p_\theta(\mathbf{x}_{\mathbf{z} \geq 1} \mid y_1, y_2) = -\sum_{t=1}^{5} \log p_\theta(x_t \mid \mathbf{x}_{\mathbf{z} < t}, y_1, y_2), \tag{13}$$

as we have analyzed in Section 4.3, both in content stream and query stream, $y_1$ and $y_2$ can be attended to every token including themselves and $x_i$ can be attended to $x_j$ if $i < j$. Additionally, $x_i$ can be attended to itself in the content stream. Therefore, the causal mask for content stream $\mathbf{M}_h$ and the causal mask for query stream $\mathbf{M}_g$ should be:

$\mathbf{M}_h =$
$$\begin{bmatrix} 0 & 0 & -\infty & -\infty & -\infty & -\infty & -\infty \\ 0 & 0 & -\infty & -\infty & -\infty & -\infty & -\infty \\ 0 & 0 & 0 & -\infty & -\infty & -\infty & -\infty \\ 0 & 0 & 0 & 0 & -\infty & -\infty & -\infty \\ 0 & 0 & 0 & 0 & 0 & -\infty & -\infty \\ 0 & 0 & 0 & 0 & 0 & 0 & -\infty \\ 0 & 0 & 0 & 0 & 0 & 0 & 0 \end{bmatrix}$$

$\mathbf{M}_g =$
$$\begin{bmatrix} 0 & 0 & -\infty & -\infty & -\infty & -\infty & -\infty \\ 0 & 0 & -\infty & -\infty & -\infty & -\infty & -\infty \\ 0 & 0 & -\infty & -\infty & -\infty & -\infty & -\infty \\ 0 & 0 & 0 & -\infty & -\infty & -\infty & -\infty \\ 0 & 0 & 0 & 0 & -\infty & -\infty & -\infty \\ 0 & 0 & 0 & 0 & 0 & -\infty & -\infty \\ 0 & 0 & 0 & 0 & 0 & 0 & -\infty \end{bmatrix}$$

$$\tag{14}$$

If we consider putting a part of $\mathbf{x}$, let us say $x_1$, into the condition and get the formulation

$$-\log p_\theta(\mathbf{x}_{\mathbf{z} \geq 2} \mid x_1, y_1, y_2) = -\sum_{t=2}^{5} \log p_\theta(x_t \mid \mathbf{x}_{\mathbf{z} < t}, x_1, y_1, y_2), \tag{15}$$

then $x_1, y_1, y_2$ can be attended to every token. The two masks should be:

$\mathbf{M}_h =$
$$\begin{bmatrix} 0 & 0 & 0 & -\infty & -\infty & -\infty & -\infty \\ 0 & 0 & 0 & -\infty & -\infty & -\infty & -\infty \\ 0 & 0 & 0 & -\infty & -\infty & -\infty & -\infty \\ 0 & 0 & 0 & 0 & -\infty & -\infty & -\infty \\ 0 & 0 & 0 & 0 & 0 & -\infty & -\infty \\ 0 & 0 & 0 & 0 & 0 & 0 & -\infty \\ 0 & 0 & 0 & 0 & 0 & 0 & 0 \end{bmatrix}$$

$\mathbf{M}_g =$
$$\begin{bmatrix} 0 & 0 & 0 & -\infty & -\infty & -\infty & -\infty \\ 0 & 0 & 0 & -\infty & -\infty & -\infty & -\infty \\ 0 & 0 & 0 & -\infty & -\infty & -\infty & -\infty \\ 0 & 0 & 0 & -\infty & -\infty & -\infty & -\infty \\ 0 & 0 & 0 & 0 & -\infty & -\infty & -\infty \\ 0 & 0 & 0 & 0 & 0 & -\infty & -\infty \\ 0 & 0 & 0 & 0 & 0 & 0 & -\infty \end{bmatrix}$$

$$\tag{16}$$

Then we consider the situation that every group $G_k$ may have more than one elements. Suppose that $G_0 = [1], G_1 = [2, 3], G_2 = [4, 5]$. That is $K = 2, n_0 = 1, n_1 = 3, n_2 = 5$. Then $x_i$ will be dependent on $x_j$ only if the serial number of $x_i$'s group is larger than that of $x_j$'s group. Therefore, the two masks should be

$\mathbf{M}_h =$
$$\begin{bmatrix} 0 & 0 & 0 & -\infty & -\infty & -\infty & -\infty \\ 0 & 0 & 0 & -\infty & -\infty & -\infty & -\infty \\ 0 & 0 & 0 & -\infty & -\infty & -\infty & -\infty \\ 0 & 0 & 0 & 0 & 0 & -\infty & -\infty \\ 0 & 0 & 0 & 0 & 0 & -\infty & -\infty \\ 0 & 0 & 0 & 0 & 0 & 0 & 0 \\ 0 & 0 & 0 & 0 & 0 & 0 & 0 \end{bmatrix}$$

$\mathbf{M}_g =$
$$\begin{bmatrix} 0 & 0 & 0 & -\infty & -\infty & -\infty & -\infty \\ 0 & 0 & 0 & -\infty & -\infty & -\infty & -\infty \\ 0 & 0 & 0 & -\infty & -\infty & -\infty & -\infty \\ 0 & 0 & 0 & -\infty & -\infty & -\infty & -\infty \\ 0 & 0 & 0 & -\infty & -\infty & -\infty & -\infty \\ 0 & 0 & 0 & 0 & 0 & -\infty & -\infty \\ 0 & 0 & 0 & 0 & 0 & -\infty & -\infty \end{bmatrix}$$

$$\tag{17}$$

At last, we consider varying the index sequence $\mathbf{z}$. Suppose we are given the index sequence $\mathbf{z} = [2, 3, 4, 5, 1]$. We only need to swap the columns and the rows of the original causal masks $\mathbf{M}_h$ and $\mathbf{M}_g$ with index sequence of $[1, 2, 3, 4, 5]$. The $z'(i)$ mentioned in Section 4.3 should be $z'(1) = 1, z'(2) = 2, z'(3) = z'(1 + 2) = 5 + 2 = 7, z'(4) = z'(2 + 2) = 1 + 2 = 3, z'(5) =$

$z'(3+2) = 2+2 = 4, z'(6) = z'(4+2) = 3+2 = 5, z'(7) = z'(5+2) = 4+2 = 6$. Notice that in this situation $x_{\mathbf{z}_1} = x_2$ is integrated into the condition. By

$$(\mathbf{M}_h^{\mathbf{z}})_{ij} = (\mathbf{M}_h)_{z'(i)z'(j)}, (\mathbf{M}_g^{\mathbf{z}})_{ij} = (\mathbf{M}_g)_{z'(i)z'(j)}, \tag{18}$$

we can obtain that

$$\mathbf{M}_h^{\mathbf{z}} = \begin{bmatrix} 0 & 0 & -\infty & 0 & -\infty & -\infty & -\infty \\ 0 & 0 & -\infty & 0 & -\infty & -\infty & -\infty \\ & & & & & & \\ 0 & 0 & 0 & 0 & 0 & 0 & 0 \\ 0 & 0 & -\infty & 0 & -\infty & -\infty & -\infty \\ 0 & 0 & -\infty & 0 & 0 & 0 & -\infty \\ 0 & 0 & -\infty & 0 & 0 & 0 & -\infty \\ 0 & 0 & 0 & 0 & 0 & 0 & 0 \end{bmatrix} \qquad \mathbf{M}_g^{\mathbf{z}} = \begin{bmatrix} 0 & 0 & -\infty & 0 & -\infty & -\infty & -\infty \\ 0 & 0 & -\infty & 0 & -\infty & -\infty & -\infty \\ & & & & & & \\ 0 & 0 & -\infty & 0 & 0 & 0 & -\infty \\ 0 & 0 & -\infty & 0 & -\infty & -\infty & -\infty \\ 0 & 0 & -\infty & 0 & -\infty & -\infty & -\infty \\ 0 & 0 & -\infty & 0 & -\infty & -\infty & -\infty \\ 0 & 0 & -\infty & 0 & 0 & 0 & -\infty \end{bmatrix} \tag{19}$$

## B  DETAILS ON SELECTING GROUP SEGMENTATION

In this section, we will illustrate the selecting details in the settings of the group segmentation proposed in Section 3.2. In the **Random Length** setting, we let $n_0, n_1, \ldots, n_{K-1}$ uniformly distributed in the range $[1, T-1]$. We implement this by first randomly permuting $[1, 2, \ldots, T-1]$, selecting the first $K$ elements of the permuted list and then sorting them in an ascending order. Here we provide the python code in PyTorch style:

```python
# Python code for selecting n_0, n_1,..., n_{K-1} in random length
    setting
import torch

def generate_random_length(K, T):
    # Create a tensor containing all integers in the range [1, T-1]
    integers = torch.arange(1, T)

    # Shuffle the tensor of integers randomly
    shuffled_integers = integers[torch.randperm(T-1)]

    # Select the first K unique integers from the shuffled tensor
    selected_integers = shuffled_integers[:K]

    # Sort the selected integers in ascending order
    sorted_integers, _ = torch.sort(selected_integers)

    return sorted_integers
```

For the **Mixed Length** setting, we first generate $p$ and get $n_{K-1} = round(p \cdot T)$. Then similarly, we select the $n_0, n_1, \ldots, n_{K-2}$ in the range $[1, n_{K-1} - 1]$. Here we provide the python code in PyTorch style:

```python
# Python code for selecting n_0, n_1,..., n_{K-1} in mixed length
    setting
import torch

def generate_mixed_length(K, T):
    # Generate a Gaussian distribution with mean 0.5 and variance
        0.1
    p = torch.normal(mean=0.5, std=0.1)

    # Calculate n_{K-1} by rounding p multiplied by T to the
        nearest integer
    n = int(torch.round(p * T))

    # Create a tensor containing all integers in the range [1, n-1]
```

```
integers = torch.arange(1, n)

# Shuffle the tensor of integers randomly
shuffled_integers = integers[torch.randperm(n-1)]

# Select the first K-1 unique integers from the shuffled tensor
selected_integers = shuffled_integers[:K-1]

# Sort the selected integers in ascending order
sorted_integers, _ = torch.sort(selected_integers)

# Concat the selected integers with n
sorted_integers = torch.cat([sorted_integers, torch.tensor(n)])

return sorted_integers
```

## C  EXPERIMENTS ON LONGER TRAINING

In this section, we conduct experiments for longer pretraining. We train our GVP-p2p and GVP-p2d for 800 epochs and warm up for 40 epochs. The other settings still keep the same. We provide linear probing and finetuning results in Table 5 and compare with other methods which also take longer training. We outperform the three MIM methods BEiT, SimMIM and MAE both in linear probing and finetuning results. It is noteworthy that our methods are on par with the autoregressive model RandSAC but we have much fewer pretraining epochs.

Table 5: Linear probing accuracy and fine-tuning accuracy of GVP-p2p and GVP-p2d pretrained with 800 epochs, along with the results of other methods.

| Method | Epochs | LP Acc.(%) | FT Acc.(%) |
|---|---|---|---|
| BEiT(dVAE) (Bao et al., 2022) | 800 | 56.7 | 83.2 |
| SimMIM (Xie et al., 2022) | 800 | 56.7 | 83.8 |
| MAE (He et al., 2022) | 800 | 65.1 | 83.6 |
| RandSAC (Hua et al., 2023) | 1600 | 68.9 | 83.9 |
| GVP-p2p | 800 | **69.6** | **84.0** |
| GVP-p2d | 800 | 67.4 | 83.8 |

## D  EXPERIMENTS ON SEMANTIC SEGMENTATION

In this section, we conduct experiments on semantic segmentation task to verify the effectiveness of our proposed approach on representation learning. We experiment on the ADE20K (Zhou et al., 2016) benchmark with 25K images and 150 semantic categories using the end-to-end framework, which follows the same setting of BEiT (Bao et al., 2022). We report the metric of mean Itersection of Union (mIoU) averaged over all semantic categories, which is adopted in many previous works (He et al., 2022; Bao et al., 2022; Hua et al., 2023). We compare our GVP-p2p and GVP-p2d with BEiT (Bao et al., 2022), MAE (He et al., 2022) and RandSAC (Hua et al., 2023). The results are shown in Table 6. When GVP-p2p outperforms the p2p methods MAE and RandSAC by 0.6, 0.2, respectively, GVP-p2d outperforms the p2d methods BEiT by 0.9, showing that our methods have better representation learning ability.

## E  EXPERIMENTS ON THE $y_{extra}$ MODULE

In this section, we conduct additional experiments to demonstrate that the incorporation of $y_{extra}$ can improve the performance on downstream tasks. The proposed $y_{extra}$ in the framework enables our approach adapt to many different tasks. The experimental settings are as follows: we feed the image into a pretrained VQGAN model to extract and pool the features of each patch. This pooled outcome represents a form of global information for the image, hereby designated as $y_{extra}$. The integration of $y_{extra}$ is applied to both the pretraining and downstream task stages in GVP-p2p variant, while

Table 6: Semantic Segmentation on ADE20K.

| Method | Pretraining Epochs | mIoU |
|---|---|---|
| BEiT (Bao et al., 2022) | 800 | 46.5 |
| MAE (He et al., 2022) | 800 | 48.1 |
| RandSAC (Hua et al., 2023) | 1600 | 48.5 |
| GVP-p2p | 200 | **48.7** |
| GVP-p2d | 200 | 47.4 |

other parameters remain constant. The results are shown in Table 7. While the VQGAN feature alone does not exhibit strong linear probing performance, its integration into the process results in a notable improvement of 4%. This outcome underscores the positive impact of incorporating $y_{extra}$.

Table 7: Experiments on using pooled features from VQGAN as $y_{extra}$. The dataset is ImageNet100. The model is pretrained with ViT-B for 200 epochs. $y_{extra}$ as pretrained model means directly using $y_{extra}$ as the pretrained features to conduct linear probing. This control group is to verify that $y_{extra}$ itself does not contain much class information.

| Pretrained Model | LP Acc. |
|---|---|
| GVP-p2p (vanilla) | 69.4 |
| $y_{extra}$ | 42.2 |
| GVP-p2p (with $y_{extra}$) | **73.1** |

## F    IMAGE RECONSTRUCTION

In this section, we conduct image reconstruction experiments, which can be visualized in Figure 2. We calculate the average LPIPS loss (Zhang et al., 2018) on the validation set of each method. Both in terms of intuitive visual perception and numerical results, our method outperforms other approaches with same epochs of pretraining.

**Discussion.** In the image reconstruction task, both conventional autoregressive methods and MIM methods exhibit shortcomings. When provided with patches at random positions within an image, conventional autoregressive methods like iGPT struggle to capture bidirectional information, limiting their ability to leverage complete information from the given patches. On the other hand, MIM methods like MAE may overlook the dependency relations between predicted patches, potentially encountering multimodality issues as discussed in (Gu et al., 2018), resulting in lower-quality generation. In contrast, our proposed methods GVP can leverage the added flexibility to enhance the generation performance. With the given patches as the condition information, we can perform the proposed group autoregressive modeling, generating predicted patches group by group. This approach will enjoy better reconstruction quality than MAE.

Table 8: Image reconstruction experiments on ImageNet validation set. The mask ratio is set to 0.75. We calculate the average LPIPS loss (Zhang et al., 2018) on the validation set of each method. Our approach outperforms the others within the same input and target configuration.

| Method | MAE | GVP-p2p | MAGE | GVP-d2d. |
|---|---|---|---|---|
| Recon. loss | 0.176 | 0.152 | 0.133 | 0.129 |

## G    EXPERIMENTS ON MS COCO

We conduct object detection experiments on the MS COCO dataset (Lin et al., 2015). We adopt ViT as the backbone of Mask-RCNN (He et al., 2018), following the setting of ViT Benchmarking (Li et al., 2021). We finetune on the MS COCO dataset for 25 epochs. We show the performance of representations learned through different self-supervised methods and supervised training. We report box AP for object detection in Table 9.

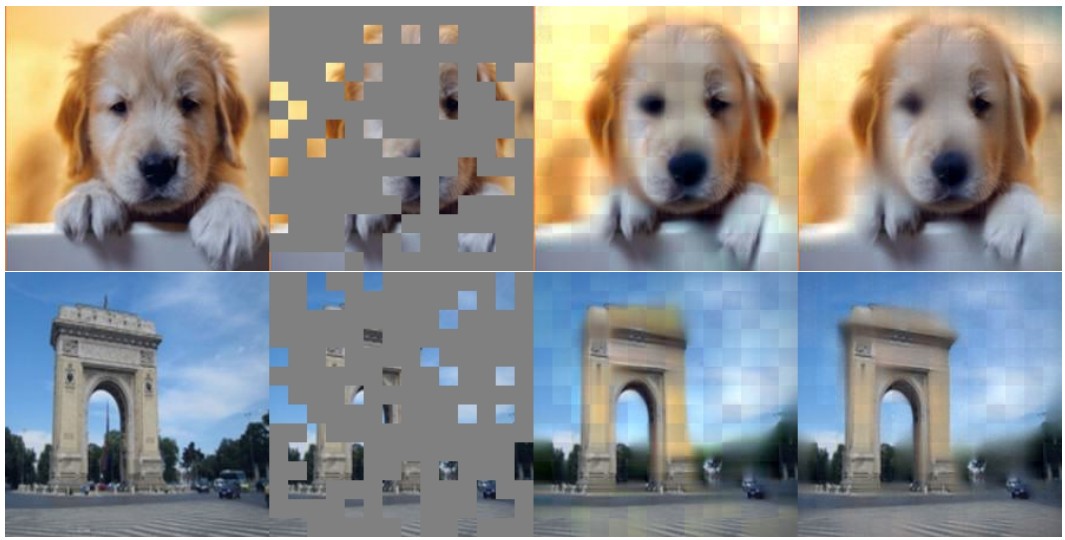

Figure 2: Image reconstruction visualization on ImageNet validation set. From left to right: Original image, mask, MAE reconstruction, GVP-p2p reconstruction. Note that GVP-p2p reconstruction has much less artifact than MAE's.

Table 9: Experiments on MS COCO dataset. The results of ViT-iGPT are borrowed from (Qi et al., 2022).

| Method | Pretraining Epochs | Finetuning Epochs | COCO-AP$^{bb}$ |
|---|---|---|---|
| ViT-iGPT | 300 | 100 | 45.5 |
| MAE | 200 | 25 | 46.2 |
| GVP-p2p (ours) | 200 | 25 | 46.6 |

We observe that our method achieves 46.6 bbox mAP with 25 epochs of finetuning. The result is better than autoregressive method iGPT and MIM method MAE. The results indicate that our method can benefit from both MIM and autoregressive methods and learn high-quality representations.

## H  ABLATION STUDY ON KEY FACTORS

We conduct ablation study by incrementally adding the modifications. The models are pretrained using ViT-B with 200 epochs on ImageNet-1k. The results are presented in Table 10. From the results, we observe that (1) Directly adding causal masks on masked modeling introduces the problem that the model is unaware of the next position it will predict, as discussed in A2. This problem will lead to deteriorated performance. Leveraging the two-stream attention mechanism mitigates this issue. (2) Without a well-designed grouping method, the performance still remains consistent with the baseline. However, with the introduction of mixed length grouping, the performance surpasses the baseline by a large margin. This underscores the significance of a well designed grouping method in the modeling.

Table 10: Ablation study on the key factor of our proposed method. The models are pretrained using ViT-B with 200 epochs on ImageNet-1k.

| Modification | Linear Acc. | Finetuning Acc. |
|---|---|---|
| Pure Masked Modeling | 52.5 | 82.8 |
| +Causal Masks | 28.4 | 78.4 |
| +Causal Masks+Two-stream Attention | 52.2 | 82.3 |
| +Causal Masks+Two-stream Attention+Mixed Length Grouping | 61.4 | 83.2 |

# I  ABLATION EXPERIMENTS WITHIN P2D SETTING

We further explore experiments using both fixed length and random length settings on p2d, with the segmentation amount $K$ set to 20. The models in these experiment are trained for 200 epochs with ViT-B on ImageNet, following the setting in Section 5. The results are presented in Table 11. The findings suggest that modeling with mixed length yields the optimal performance, aligning with the outcomes observed in the p2p experiments.

Table 11: Ablation study on different segmentation settings in the p2d setting.

| Segmentation setting | LP Acc. (%) | FT Acc. (%) |
|---|---|---|
| Fixed Length | 53.9 | 82.6 |
| Random Length | 56.2 | 82.6 |
| Mixed Length | 58.3 | 83.4 |

