# OpenReview forum: "Bridging Autoregressive and Masked Modeling for Enhanced Visual Representation Learning"
_ICLR.cc/2024/Conference — Submitted to ICLR 2024_

### Official Review · Reviewer_BXi8 · 2023-10-16

**Soundness:** 3 good
**Presentation:** 3 good
**Contribution:** 2 fair
**Rating:** 5
**Confidence:** 5

**Summary:**

This paper proposes a method called "Generative Visual Pretraining (GVP)" aimed at exploring the potential of autoregressive models in the field of computer vision. Autoregressive models have demonstrated remarkable performance in natural language processing, but they have not been fully explored in computer vision, thus there are still some key challenges.

By probabilistically analyzing the methods of autoregressive and autoencoding mask modeling, it is found that the two can complement each other. Based on this, a general formula and modeling framework are proposed to combine the advantages of both methods, named "Generative Visual Pretraining (GVP)". Their unified probability framework allows for the simultaneous implementation of different training strategies, including mask modeling and autoregressive modeling. Their framework can adapt to various downstream tasks and outperforms existing methods in multiple benchmark tests, including linear probing, fine-tuning, and transfer learning.

**Strengths:**

Comprehensive Framework: The paper proposes a unified probabilistic framework that combines autoregressive modeling and masked modeling methods, enabling them to simultaneously implement different training strategies, including masked modeling and autoregressive modeling.

Performance Improvement in Tasks: The authors validate the effectiveness of their framework in multiple benchmark tests, including linear detection, fine-tuning, and transfer learning. They demonstrate that the framework outperforms existing methods in these tasks.

Flexible Generation Capability: Due to the integration of the advantages of autoregressive and masked modeling methods, the framework can provide improved classification ability and flexible generation capability.

Mathematical Analysis: The article analyzes the relationship between current autoencoding methods and autoregressive methods from a probabilistic perspective, and proposes a group-based autoregressive probability framework that allows modeling and flexible adjustment of task complexity in any order while modeling strong dependencies between tokens.

**Weaknesses:**

The author introduces a new framework that performs well in multiple benchmark tests. However, the paper lacks an in-depth discussion of the framework's semantic effectiveness in visual multi-object scenes. Although the author validates the effectiveness of the segmentation task ADE20K, there is insufficient validation on other detection and segmentation tasks. This limits the applicability of the method in real visual scenarios.

Furthermore, the method is not compared with last year's SAIM paper[1]. SAIM and RandSAC are contemporaneous works that explore self-supervised approaches using autoregressive methods. The autoregressive self-supervised method has also shown good performance on downstream tasks. As far as I know, RandSAC has explored group-level reasoning. How can we demonstrate the advantages of this paper's method compared to SAIM's patch-based autoregressive and RandSAC's segment-based approaches? Is it possible that your method is just a combination of these two methods?

Additionally, this article does not describe the efficiency of the method. We need to know if the method significantly delays training time, thereby reducing training efficiency. Moreover, the author uses a large batch size, which may be to obtain better performance. However, I hope the author can provide a more detailed analysis of this design choice.

[1] Qi, Yu and Yang, Fan and Zhu, Yousong and Liu, Yufei and Wu, Liwei and Zhao, Rui and Li, Wei. Exploring stochastic autoregressive image modeling for visual representation, 2022.

**Questions:**

The current paper does not provide results for the multi-object visual scene detection task on COCO. It is recommended to include experimental results for downstream validation tasks related to object detection.

Please explain why only the mixed results perform the best in the autoregressive analysis. How do the fixed length and random approaches based on p2d and d2d perform?

Please provide the inference time and training time.

Is the GPU usage higher for this method compared to the RandSAC and SAIM methods?

---

> ### Author Response · Authors · 2023-11-20
> **Reply to Reviewer BXi8 (1/2)**
>
> We thank Reviewer BXi8 for careful reading and detailed comments. We address your concerns in the following points:
>
> ---
>
> Q1. The current paper does not provide results for detection task on COCO. It is recommended to include experimental results for downstream validation tasks related to object detection.
>
> A1. Following your suggestion, we conduct object detection experiments on the MS COCO dataset [1]. We adopt ViT as the backbone of Mask-RCNN [2], following the architecture of ViT Benchmarking [3]. Due to the limited time, we only finetune the model for 25 epochs. We show the performance of representations learned through different self-supervised methods and supervised training. We report box AP for object detection, where the result of ViT-iGPT is borrowed from [4].
>
> |Method|Pretraining Epochs|Finetuning Epochs|COCO-AP$^{bb}$|
> |---|---|---|---|
> |ViT-iGPT|300|100|45.5|
> |MAE|200|25|46.2|
> |GVP-p2p (ours)|200|25|46.6|
>
> We observe that our method achieves 46.6 bbox mAP with 25 epochs of finetuning, which is better than autoregressive method iGPT and MIM method MAE. The results indicate that our method can benefit from both MIM and autoregressive methods and learn high-quality representations. We have added this experiment in Appendix G in the revision.
>
> [1] Lin et al. Microsoft COCO: Common Objects in Context, Arxiv 1405.0312.
>
> [2] He et al. Mask R-CNN, Arxiv 1703.06870.
>
> [3] Li et al. Benchmarking Detection Transfer Learning with Vision Transformers, Arxiv 2111.11429
>
> [4] Qi et al. Exploring Stochastic Autoregressive Image Modeling for Visual Representation, AAAI 2023.
>
> ---
>
> Q2. The method should be compared with last year's SAIM paper, which is a concurrent work with RandSAC. Is it possible that your method is just a combination of these two methods?
>
> A2. Thanks for bringing up this previous work! SAIM is also an excellent work building an autoregressive model within the CV field. However, SAIM only considers the patch-level autoregressive modeling and neglect the advantages brought from MIM. In contrast, we provide a unified perspective and effectively fuse the targets of both paradigms. **Our proposed GVP is not a mere combination of RandSAC and SAIM; rather, GVP represents a more generalized and effective modeling ways of images**. Here, we list three points on the comparison among our proposed method GVP, SAIM and RandSAC below:
>
> 1. **Unified perspective on the inter-patch dependency relationship**: Our formulation provides a unified perspective on the dependency relationships between MIM and autoregressive model. Based on this perspective, we formulate the fused targets, which are aimed to corporate the merits of MIM and autoregressive tasks. Therefore, we can benefit from both pretraining paradigms. On the other hand, **RandSAC and SAIM just treat their models as an autoregressive model and neglect the benefits that MIM may bring.** Indeed, our proposed GVP-p2p has better performance in classification tasks as shown in the table below:
>
> |Model|Epochs|Linear Probing Acc.|Finetuning Acc.|
> |---|---|---|---|
> |RandSAC|1600|68.9|83.9|
> |SAIM|800|62.5|83.9|
> |GVP-p2p|800|69.6|84.0|
>
> 2. **Additional module $y_{extra}$**: Another improvement involves the introduction of the concept of $y_{extra}$ within our framework, enhancing the adaptability of our approach across diverse tasks. To substantiate this assertion, we conduct additional experiments to demonstrate that the incorporation of $y_{extra}$ can improve the performance on downstream tasks. The experimental settings are as follows: we feed the image into a pretrained VQGAN model to extract and pool the features of each patch. This pooled outcome represents a form of global information for the image, hereby designated as $y_{extra}$. The integration of $y_{extra}$ is applied to both the pretraining and downstream task stages in GVP-p2p variant, while other parameters remain constant. The results are shown in Appendix E. While the VQGAN feature alone does not exhibit strong linear probing performance, its integration into the process results in a notable improvement of 4%. This outcome underscores the positive impact of $y_{extra}$.
>
> 3. **Form of Input and Prediction Target**: In our paper, we explore diverse settings for the form of input and prediction Target, including pixel to pixel (p2p), pixel to discrete (p2d) and dicrete to discrete (d2d). **This exploration serves to highlight the flexibility and generality of our proposed methods.** In contrast, SAIM discusses about the Gaussian smoothing applied to the prediction target, which enhances the model's capacity to capture information about the low-frequency levels of the image. RandSAC only adopts the raw pixel of the patch to be the reconstruction target. **These two works overlook the diversity of input and prediction forms in the field of representation learning and, as a result, are not general enough.**
>
> We have cited SAIM and added more discussions in Section 2 in the revision.
>
> ---

---

> > ### Comment · Area_Chair_kd4E · 2023-12-04
> >
> > Have the authors' responses and additional results addressed your concerns? Given the mixed reviews, with two 'borderline below acceptance' and two 'borderline above acceptance' ratings, your detailed feedback at this juncture is crucial.

---

> ### Author Response · Authors · 2023-11-20
> **Reply to Reviewer BXi8 (2/2)**
>
> Q3. This article should describe the efficiency of the method, including the inference time and training time. Moreover, the author uses a large batch size, which may be to obtain better performance. However, I hope the author can provide a more detailed analysis of this design choice.
>
> A3. We conduct experiments using 8 nodes, each of which contains 8 A100 GPU. The comparison of pretraining time on ImageNet with ViT-B for 200 epochs is listed below:
>
> | Method | Pretraining time (hours) |
> | -------- | -------- |
> | MAE (p2p)   | 11.7 |
> | BEiT (p2d)  | 13.2 |
> | MAGE (d2d)  | 13.1 |
> | GVP-p2p | 15.8|
> | GVP-p2d | 16.7|
> | GVP-d2d | 16.8|
>
> In comparison to the baseline methods, our approach requires approximately 30% additional time within each input and target configuration. This increment in time is acceptable given the corresponding enhancement in performance.
>
> Regarding inference time, the incorporation of the causal mask does not introduce any additional inference time overhead. Consequently, the inference time remains consistent with methods using ViT as the architecture, such as MAE and BEiT, as shown in the table below:
>
> |Method|Inference Time using the ViT-B backbone over ImageNet-1K|
> |---|---|
> |MAE|63.1s|
> |GVP-p2p|63.1s|
>
> Regarding the batch size, the batch size of 4096 is consistent with several baselines such as MAE and RandSAC. We just follow this default batch size and do not mean to choose a large batch size.
>
> ---
>
> Q4. Please explain why only the mixed results perform the best in the autoregressive analysis. How do the fixed length and random approaches based on p2d and d2d perform?
>
> A4. The proposed mixed length setting combines the challenging prediction tasks inherent to MAE, random task complexity from MAGE, and the diverse inter-patch dependency relationships from AR. Therefore, it will outperform the other two settings that lack challenging prediction tasks.
>
> We further explore experiments using both fixed length and random length settings on p2d, with the segmentation amount $K$ set to 20. The models in these experiment are trained for 200 epochs with ViT-B on ImageNet, following the setting in our paper. The results are presented below:
>
> |Segmentation setting|LP Acc.(%)|FT Acc.(%)|
> |---|---|---|
> |Fixed Length|53.9|82.6|
> |Random Length|56.2|82.6|
> |Mixed Length|58.3|83.4|
>
> The findings suggest that modeling with mixed length yields the optimal performance, aligning with the outcomes observed in the p2p experiments. We have added this experiment in Appendix I. Due to time constraints, the ablation experiments for d2d will be conducted in the future, where we expect that the results will align with those observed in p2d.
>
>
> ---
>
> Q5. Is the GPU usage higher for this method compared to the RandSAC and SAIM methods?
>
> A5. As our proposed method employs an architecture requiring two streams of forward propagation, aligning with the design of RandSAC and SAIM methods, the GPU usage remains consistent with that of RandSAC and SAIM.
>
> ---
>
> Thanks for your comments and hope our answers could address your concerns. Please let us know if you have additional questions.

---

> ### Author Response · Authors · 2023-11-23
>
> Dear Reviewer BXi8,
>
> We have carefully prepared a detailed response to address each of your questions. Would you please take a look and let us know whether you find it satisfactory? Furthermore, we have made revisions to our paper based on your suggestions. Your detailed suggestions have been invaluable in improving our paper. We respectfully suggest that you could re-evaluate our work with the updated explanations and results.
>
> Thanks! Have a great day!
>
> Authors

---

### Official Review · Reviewer_N26k · 2023-10-24

**Soundness:** 3 good
**Presentation:** 2 fair
**Contribution:** 3 good
**Rating:** 6
**Confidence:** 4

**Summary:**

The paper explores the idea from XL-net (generalized auto-regressive modeling) in NLP that can efficiently implement arbitrary ordering to self-supervised visual representation learning (which is currently dominated by the idea of masked modeling). The paper provides a perspective that bridges the two type of modeling methods (all using visible data to predict hidden ones), and borrows the idea from XL-net to get the best of two worlds. Experiments are conducted on both pixel input/output, and tokenized input/output, and some improvements are shown (in Table 1). Analysis is mainly on the claim that iGPT is too restricted in using a fixed ordering (random ordering helps), and on how to do sequential prediction in groups.

**Strengths:**

+ The clear write-up that bridges the gap between masked modeling and autoregressive modeling is solid and very easy to follow.
+ I appreciate the effort of applying the idea from XL-net to see the effect. Even further, the work explored different inputs/outputs, different ways to evaluate the representation (linear vs. fine-tune), etc, which I highly appreciate.

**Weaknesses:**

- I understand computation could be a problem, but would like to see more solid comparisons. E.g., the main table (Table 1) used a pre-training epoch of 200. At this stage, it is unclear whether the pre-training is just worse in upper bound, or it is just some pre-training converges faster. Plus, it would be much more fair to me if the comparison is done based on the "overall compute" or "overall data" spent on pre-training, rather than simply put number of epochs there. For example, the default number of epochs used in MAE is 800-ep, this is when the model has sufficiently converged. And even then, from 800 to 1600 there is still notable improvements in the final table. Comparing with MAE at 200-epoch is not the fair-most comparison.
- (minor) I think Eq 3 has a typo that z should be greater or equal to m, instead of t.

**Questions:**

- Is there a planned code release? I guess from XL-net it is fine but would be great to promote open-sourcing here.

---

> ### Author Response · Authors · 2023-11-20
> **Reply to Reviewer N26k**
>
> We thank Reviewer N26k for the appreciation on our work and detailed comments. We address your concerns in the following points:
>
> ---
>
> Q1. I would like to see more solid experimental results such as longer pretraining.
>
> A1. We have included experiments with 800 epochs pretraining on ImageNet in Appendix C. We will also show the results in the following table:
> |Method|Epochs|LP Acc.|FT Acc.|
> |---|---|---|---|
> |BEiT|800|56.7|83.2|
> |SimMIM|800|56.7|83.8|
> |MAE|800|65.1|83.6|
> |RandSAC|1600|68.9|83.9|
> |GVP-p2p|800|**69.6**|**84.0**|
> |GVP-p2d|800|67.4|83.8|
>
> We outperform the three MIM methods BEiT, SimMIM and MAE both in linear probing and finetuning results. It is noteworthy that our methods are on par with the autoregressive model RandSAC but we have much fewer pretraining epochs (Ours 800 epochs v.s. RandSAC 1600 epochs).
>
> ---
>
> Q2. Equation (3) has a typo that $z$ should be greater or equal to $m$, instead of $t$.
>
> A2. Thanks for pointing it out. We have corrected it in the revision.
>
> ---
>
> Q3. Is there a planned code release?
>
> A3. Sure, we will definitly release the code upon acceptance.
>
> ---
>
> Thanks for your comments and hope our answers could address your concerns. Please let us know if you have additional questions.

---

> ### Author Response · Authors · 2023-11-23
>
> Dear Reviewer N26k,
>
> We have carefully prepared a detailed response to address each of your questions. Would you please take a look and let us know whether you find it satisfactory? Furthermore, we have made revisions to our paper based on your suggestions. Your detailed suggestions have been invaluable in improving our paper. We respectfully suggest that you could re-evaluate our work with the updated explanations and results.
>
> Thanks! Have a great day!
>
> Authors

---

> > ### Comment · Reviewer_N26k · 2023-11-23
> >
> > Thanks, I acknowledge I have read the rebuttal, and find them to be helpful. I am already on the acceptance side and only had minor issues which have been addressed, I am going to keep my score. On the other hand, I feel it's better to move some results (e.g., longer training) to the main paper for the revision.

---

### Official Review · Reviewer_mLNs · 2023-10-30

**Soundness:** 2 fair
**Presentation:** 2 fair
**Contribution:** 2 fair
**Rating:** 6
**Confidence:** 5

**Summary:**

This paper found that autoregressive modeling and masked modeling can complement each other for visual representation learning. The authors proposed a general framework that combines both methods. Experiments on several vision benchmarks show that this framework performs better than previous methods.

**Strengths:**

This paper demonstrates something interesting to me. It proposed a general framework for autoregressive modeling and masked modeling paradigms, which may help to better formulate the problem. In doing so, it also revealed a difference lies in the sequential dependency of autoregressive modeling and independence assumption of masked modeling, which I think may serve as a useful starting point to study generative visual pre-training.

**Weaknesses:**

While the innovation is good, I found some perspectives of the paper are not sound enough to convey true insights. Details are listed below.

1.	The motivation is not clearly illustrated. The paper claimed that “autoregressive models and masked modeling can complement each other” in Paragraph 3 of the introduction. However, I’m not aware of why “naturally generate sequences based on a specific masking pattern” is a good property. In fact, as the authors have stated, predicting autoregressively is not a natural choice in computer vision. Then, what’s the benefit of combining autoregressive models?

2.	The design of two-stream attention is not well motivated. Previous methods like MAE use one-stream attention. The proposed framework does not put a restriction on the two-stream attention mechanism. What is the purpose of introducing two-stream attention? On the other hand, two-stream attention is not well presented. What’s the differences between content stream and query stream? Why the causal masks are different for these two streams?

3.	Lack of ablation studies. Two ablation studies in the paper (patch ordering, analysis on auto-regressivity) both demonstrated that fixed pattern is not helpful for representation learning. However, these properties are consistent with masked modeling, but instead is not possessed by autoregressive modeling. As I have mentioned in Weakness 1, these experiments still do not tell us exactly what is the benefit of using autoregressive models.
The main results show that the general framework performs better. Ablation studies should be responsible for telling why the performance is better. In order to do this, the authors need rigorous comparison experiments, especially with masked modeling paradigm. For example, experiments can begin with a pure masked modeling version (no grouping, no causal masks, no two-stream attention), and gradually add these modifications step by step to show what is key factor that contributes to the performance gain.

**Questions:**

1.	Notation is chaotic.

*	Equation (2) , $x$ is of different styles and inconsistent subscripts. Notation $m$ is not introduced.

*	Page 6, Section 4.2, Para 2, Line 4, $M$ is not introduced.

2.	What is the group segmentation distribution during training?

3.	Do you have the results on COCO detection task?

---

> ### Author Response · Authors · 2023-11-20
> **Reply to Reviewer mLNs (1/2)**
>
> We thank Reviewer mLNs for careful reading and detailed comments. We address your concerns in the following points:
>
> ---
>
> Q1. The motivation is not clearly illustrated. The paper claimed that “autoregressive models and masked modeling can complement each other”. However, I’m not aware of why “naturally generate sequences based on a specific masking pattern” is a good property. In fact, as the authors have stated, predicting autoregressively is not a natural choice in computer vision. Then, what’s the benefit of combining autoregressive models?
>
> A1. We now understand your point on clarifying the benefit of autoregressive model. Indeed, as you mentioned, conventional autoregressive methods such as iGPT, face a challenge in that the fixed raster order they adopt is not a natural choice. This limitation makes the model incapable of effectively capturing bidirectional information. However, **autoregressive models enjoy the advantage of effectively capturing intricate dependency relationships between tokens.** This stands in contrast to the approach in MIM, where the predicted tokens are treated independently. As discussed in Section 3, this independence in token prediction can result in lower generation quality.
>
> Given the pros and cons of the two paradigms, **our motivation is to create a synergy between them by developing a novel modeling method.** We hope that the introduction of modeling intricate dependency by autoregressive modeling will assist MIM in learning more inter-image information and enhance its performance in generation tasks. In fact, by incoporating autoregressive modeling, GVP demonstrates improved image reconstruction ability compared to MAE, as evidenced in Appendix F. **We clarify the advantage of autoregressive model and make the motivation clearer in the introduction part in the revision.**
>
> ---
>
> Q2. The design of two-stream attention is not well motivated. Previous methods like MAE use one-stream attention. The proposed framework does not put a restriction on the two-stream attention mechanism. What is the purpose of introducing two-stream attention? On the other hand, two-stream attention is not well presented. What’s the differences between content stream and query stream? Why the causal masks are different for these two streams?
>
> A2. We understand your concern on the motivation and demonstration on the two-stream attention. We will clarify the motivation and present some details of two-stream attention both here and in the revision.
>
> **The purpose of introducing two-stream attention.** As has been discussed in the backbone network part in Section 4.1, we are not capable of modeling with arbitrary index sequence in an autoregressive way if we use a single stream in ViT with causal masks. This is because of two requirements that are contradictory in a standard ViT architecture, which have been discussed in XLNet [1]: (1) to predict the token $x_{z_t}$, the network output in the $z_t$ position in one layer $g_{\theta}(x_{\mathbf{z}<t}, z_t)$ should only use the position $z_t$ and not the content $x_{z_t}$, (2) to predict the other tokens $x_{z_j}$ with $j > t$, $g_{\theta}(x_{\mathbf{z}<t}, z_t)$ should also encode the content $x_{z_t}$ to provide full contextual information. Therefore, it is not enough with only one stream. Since the two-stream attention proposed by XLNet facing the same problem works well, we adopt the two-stream attention proposed to solve this contradiction.
>
> **Introduction of the two-stream attention.** The differences between content stream and query stream are shown below:
> * The content representation $h_{\theta}(x_{\mathbf{z}\leq t})$ serves a similar role to the standard hidden states in ViT. This representation encodes both the condition $x_{\mathbf{z}< t}$ and $x_{z_t}$ itself. Therefore, the attention masks for the content stream should ensure that $x_{z_t}$ can be attended to itself. The content stream satisfies the second requirement in the above discussion.
> * The query representation $g_{\theta}(x_{z<t}; z_t)$ only has access to the condition $x_{\mathbf{z}< t}$ and the position $z_t$, but not the content $x_{z_t}$. Therefore, the attention masks for the query stream should ensure that $x_{z_t}$ cannot be attended to itself. The query content stream satisfies the first requirement in the above discussion.
>
> Therefore, the query, key and value in the content stream attention block are all from itself to fully encode contextual information. On the other hand, the key and value in the query stream attention block are from the content stream but the query is from the query stream itself to provide positional information. At last, the output from the query stream is seemed as the prediction result.
>
> We have added these explanations in Section 4.1 and 4.2 in the revision to achieve better presentation.
>
> [1] Yang et al. XLNet: Generalized Autoregressive Pretraining for Language Understanding, Arxiv 1906.08237.
>
> ---

---

> ### Author Response · Authors · 2023-11-20
> **Reply to Reviewer mLNs (2/2)**
>
> Q3. The authors need rigorous comparison experiments, especially with masked modeling paradigm. For example, experiments can begin with a pure masked modeling version (no grouping, no causal masks, no two-stream attention), and gradually add these modifications step by step to show what is key factor that contributes to the performance gain.
>
> A3. Thanks for the suggestion! Following your suggestion, we conduct ablation study by incrementally adding the modifications. The models are pretrained using ViT-B with 200 epochs on ImageNet-1k. The results are presented in the table below:
> |Modification|Linear Accuracy|Finetuning Accuracy|
> |---|---|---|
> |Pure Masked Modeling|52.5|82.8|
> |+Causal Masks|28.4|78.4|
> |+Causal Masks+Two-stream Attention|52.2|82.3|
> |+Causal Masks+Two-stream Attention+Mixed Length Grouping|61.4|83.2|
>
> From the results, we observe that (1) Directly adding causal masks on masked modeling introduces the problem that the model is unaware of the next position it will predict, as discussed in A2. This problem will lead to deteriorated performance. Leveraging the two-stream attention mechanism mitigates this issue. (2) Without a well-designed grouping method, the performance still remains consistent with the baseline. However, with the introduction of mixed length grouping, the performance surpasses the baseline by a large margin. This underscores the significance of a well designed grouping method in the modeling. We have added this ablation study in Appendix H.
>
> ---
>
> Q4. About the notation: (1) In Equation (2), $x$ is of different styles and inconsistent subscripts. Notation $m$ is not introduced. (2) Page 6, Section 4.2, Para 2, Line 4, $M$ is not introduced.
>
> A4. Thanks for point them out! We apologize for any confusion caused by these typos. They have been fixed in the revised version.
>
> ---
>
> Q5. What is the group segmentation distribution during training?
>
> A5. The group segmentation setting is chosen to be the mixed length setting with $K=20$ during training in the experiments shown in Table 1,2 and 3. Table 4 exhibits the performance under different group segmentation distributions.
>
> ---
>
> Q6. Do you have the results on COCO detection task?
>
> A6. We conduct object detection experiments on the MS COCO dataset [2]. We adopt ViT as the backbone of Mask-RCNN [3], following the architecture of ViT Benchmarking [4]. Due to the limited time, we only finetune the model for 25 epochs. We show the performance of representations learned through different self-supervised methods and supervised training. We report box AP for object detection, where the result of ViT-iGPT is borrowed from [5].
>
> |Method|Pretraining Epochs|Finetuning Epochs|COCO-AP$^{bb}$|
> |---|---|---|---|
> |ViT-iGPT|300|100|45.5|
> |MAE|200|25|46.2|
> |GVP-p2p (ours)|200|25|46.6|
>
> We observe that our method achieves 46.6 bbox mAP with 25 epochs of finetuning, which is better than autoregressive method iGPT and MIM method MAE. The results indicate that our method can benefit from both MIM and autoregressive methods and learn high-quality representations. We have added this experiment in Appendix G in the revision.
>
>
> [2] Lin et al. Microsoft COCO: Common Objects in Context, Arxiv 1405.0312.
>
> [3] He et al. Mask R-CNN, Arxiv 1703.06870.
>
> [4] Li et al. Benchmarking Detection Transfer Learning with Vision Transformers, Arxiv 2111.11429
>
> [5] Qi et al. Exploring Stochastic Autoregressive Image Modeling for Visual Representation, AAAI 2023.
>
> ---
>
> Thanks for your comments and hope our answers could address your concerns. Please let us know if you have additional questions.

---

> ### Author Response · Authors · 2023-11-23
>
> Dear Reviewer mLNs,
>
> We have carefully prepared a detailed response to address each of your questions. Would you please take a look and let us know whether you find it satisfactory? Furthermore, we have made revisions to our paper based on your suggestions. Your detailed suggestions have been invaluable in improving our paper. We respectfully suggest that you could re-evaluate our work with the updated explanations and results.
>
> Thanks! Have a great day!
>
> Authors

---

> ### Comment · Reviewer_mLNs · 2023-11-23
>
> I appreciate the author's response to address most of my concerns. Therefore, I have raised my score to 6. Still, I suggest the authors to think more deeply about the benefits of autoregressive modeling in vision domain and polish their paper. It is good to explore the insights from XLNet, but the authors should take the paper's topic into consideration (GVP mainly focuses on recognition tasks, and generation ability is not the main topic; also, pay attention to the citation from XLNet: $g_\theta(x_{\mathrm{z}<t},z_t)$ has not been introduced before using).

---

### Official Review · Reviewer_kXQF · 2023-10-31

**Soundness:** 2 fair
**Presentation:** 2 fair
**Contribution:** 2 fair
**Rating:** 5
**Confidence:** 3

**Summary:**

This work explores self-supervised training of visual representations with transformers models based on autoregressive and masked modelling objectives. At training time, content and query tokens and their attention masks are set up to represent a random permutation of input tokens that are predicted K at time via the query tokens.

The work claims the obtained representations outperform existing methods in linear-probing, finetune in ImageNet-1K and transfer-learning to 9 downstream tasks. It compares against MIM and AR approaches as well as both when using discrete and pixel space for input and outputs.

**Strengths:**

The presented idea of how to extend pretraining to capture both AR and MIM setup is simple, if it overperforms existing methods or its inference flexibility can be shown to be usable to new setups, I would consider it to be a valuable contribution. IMO as it currently stands it is misleading (see weaknesses).

It’s good to see the evaluation against many existing methods and cover both pixel and discrete space.

**Weaknesses:**

The presented method appears to cost at least 2x to train to alternative methods (due to processing tokens once for content and once for query). By failing to present and discuss compute cost, the comparison with existing works in the main text does not provide sufficient evidence to claim outperforming them. Besides compute the memory requirements also change and may make this method less interesting than existing approaches.

Given the above and that the methods don't differ so much in representation accuracy, it begs the question on why the extra complexity. This work provides little discussion/results/ideas on how the added sampling flexibility can be use downstream and it reads more like: "here is another (more complex) way to obtain very similar image representations".

**Questions:**

What is the training cost of the presented method vs existing methods?

What is the source for the numbers in table 1? How were hyper parameters tuned for those?

Is the added flexibility, on how these networks can be sampled, usable for downstream tasks?

---

> ### Author Response · Authors · 2023-11-20
> **Reply to Reviewer kXQF (1/2)**
>
> We thank Reviewer kXQF for careful reading and detailed comments. We address your concerns in the following points:
>
> ---
>
> Q1. What is the training cost of the presented method vs existing methods?
>
> A1. **Pretraining Time.** We conduct experiments using 8 nodes, each of which contains 8 A100 GPU. The comparison of pretraining time on ImageNet with ViT-B for 200 epochs is listed below:
>
> | Method | Pretraining time (hours) |
> | -------- | -------- |
> | MAE (p2p)   | 11.7 |
> | BEiT (p2d)  | 13.2 |
> | MAGE (d2d)  | 13.1 |
> | GVP-p2p | 15.8|
> | GVP-p2d | 16.7|
> | GVP-d2d | 16.8|
>
> In comparison to the baseline methods, our approach requires approximately 30% additional time within each input and target configuration. This increment in time is acceptable given the corresponding enhancement in performance.
>
> **Inference Time.** Regarding inference time, the incorporation of the causal mask does not introduce any additional inference time overhead. Consequently, the inference time remains consistent with methods using ViT as the architecture, such as MAE and BEiT.
>
> **Memory.** As for the memory, the memory cost is consistent with existing autoregressive methods like RandSAC [1] and SAIM [2], which also need two streams in the forward propagation.
>
> [1] Hua et al. Self-supervision through Random Segments with Autoregressive Coding (RandSAC), CVPR 2023.
>
> [2] Qi et al. Exploring Stochastic Autoregressive Image Modeling for Visual Representation, AAAI 2023.
>
> ---
>
> Q2. What is the source for the numbers in table 1? How were hyper parameters tuned for those?
>
> A2. The results of iGPT and MaskGIT are from their original papers. For the remaining methods, we implement them by ourselves using their official code. The learning rate is searched within the set $\\{0.5x, x, 2x\\}$, where $x$ represents the default learning rate provided by the respective methods. Our obtained results align consistently with those reported in prior studies [3][4][5].
>
> [3] Li et al. MAGE: MAsked Generative Encoder to Unify Representation Learning and Image Synthesis, CVPR 2023.
>
> [4] Zhang et al. How Mask Matters: Towards Theoretical Understandings of Masked Autoencoders, NIPS 2022.
>
> [5] Xie et al. Masked Frequency Modeling for Self-Supervised Visual Pre-Training. ICLR 2023.
>
> ---

---

> ### Author Response · Authors · 2023-11-20
> **Reply to Reviewer kXQF (2/2)**
>
> Q3. This work provides little discussion on how the added sampling flexibility can be use downstream and it reads more like: "here is another (more complex) way to obtain very similar image representations". Is the added flexibility, on how these networks can be sampled, usable for downstream tasks?
>
> A3. Indeed, **the added flexibility on the network sampling is usable for downstream tasks.** Our proposed flexible network design integrates the modeling paradigm from both mask modeling and autoregressive tasks, which is proved to be beneficial to various downstream tasks in the field of NLP [6]. As discussed in [6], the flexible attention masks enable the pretrained model suitable for both classification task and generation task. We believe that the flexible modeling patterns among patches can also bring benefits to various downstream tasks, which can be demonstrated by the following experimental results:
>
> 1. In the classification task, our proposed GVP achieves better performance than both MIM and autoregressive baselines that only take one modeling paradigm. Additionally, as one part of the model, the flexible attention mask ensures the alignment between the input distribution in the pretraining stage and the linear probing stage, which is helpful to the generalization (linear accuracy on ImageNet: with attention mask 61.5 v.s. without attention mask 56.2).
>
> 2. In the image reconstruction task, both conventional autoregressive methods and MIM methods exhibit shortcomings. When provided with patches at random positions within an image, conventional autoregressive methods like iGPT struggle to capture bidirectional information, limiting their ability to leverage complete information from the given patches. On the other hand, MIM methods like MAE may overlook the dependency relations between predicted patches, potentially encountering multimodality issues as discussed in [7], resulting in lower-quality generation. In contrast, our proposed methods GVP can leverage the added flexibility to enhance the generation performance. With the given patches as the condition information, we can perform the proposed group autoregressive modeling, generating predicted patches group by group. This approach will enjoy better reconstruction quality than MAE. Details can be viewed in Appendix F.
>
> Following your suggestions, we have added the above discussion on how the flexibility is usable for downstream tasks in Section 5 and Appendix F. We hope that this discussion will demonstrate the superiority of our proposed framework on benefitting downstream tasks.
>
> [6] Dong et al. Unified Language Model Pre-training for Natural Language Understanding and Generation, NeurIPS 2019.
>
> [7] Gu et al. Non-autoregressive neural machine translation. ICLR 2018.
>
>
> ---
>
> Thanks for your comments and hope our answers could address your concerns. Please let us know if you have additional questions.

---

> ### Author Response · Authors · 2023-11-23
>
> Dear Reviewer kXQF,
>
> We have carefully prepared a detailed response to address each of your questions. Would you please take a look and let us know whether you find it satisfactory?  Furthermore, we have made revisions to our paper based on your suggestions. Your detailed suggestions have been invaluable in improving our paper. We respectfully suggest that you could re-evaluate our work with the updated explanations and results.
>
> Thanks! Have a great day!
>
> Authors

---

> > ### Comment · Area_Chair_kd4E · 2023-12-04
> >
> > Have the authors' responses and additional results addressed your concerns? Given the mixed reviews, with two 'borderline below acceptance' and two 'borderline above acceptance' ratings, your detailed feedback at this juncture is crucial.

---

> ### Comment · Reviewer_kXQF · 2023-12-05
>
> I have read the authors replies and reread the new version of the submission and it does not addresses my raised concerns with enough to quality to modify my rating.
>
> 1) I retain the claim that "By failing to present and discuss compute cost, the comparison with existing works in the main text does not provide sufficient evidence to claim outperforming them". The added information on training time does not addresses this... if any it shows that the other methods could have trained at a 2x the batch size of GVP (and therefore 2x more epochs at the same number of tokens being processed by the transformer) and only increase its training time by 30%.
>
> 2) I asked for this line of work so that one could have a reason to advocate in favor of this work even without the clear evidence asked in (1).  The author justifications was just that it gets better numbers, but without (1) resolved I can't trust that argument.

---

### Meta-Review · Area_Chair_kd4E · 2023-12-11

**Metareview:**

The paper presents Generative Visual Pretraining (GVP), which combines autoregressive and masked modeling objectives to enhance self-supervised training of visual representations with transformer models. The work claims that the obtained representations outperform existing methods in linear-probing, finetuning on ImageNet-1K, and transfer learning to downstream tasks.

The paper received mixed feadback and bordline scores (6, 6, 5, 5). The reviewers appreciated the clear presentation of the paper, the idea of combining autoregressive and masked modeling, and comprehensive experimentation. However, the majority of reviewers expressed concerns about the increased training cost, added complexity, and the selection of downstream tasks to effectively showcase the method's capabilities. In particular, Reviewer kXQF highlighted the importance of a fair comparison regarding training costs, including batch size, token budget, and throughput, which is currently lacking in the paper.

After a thorough examination of all reviews, author's responses, and discussions, the AC believe the paper is not ready for acceptance in its current form.

**Justification For Why Not Higher Score:**

I didn't give it higher score mainly because 1) there are concerns not resolved in the rebuttal, and 2) the paper got borderline scores.

**Justification For Why Not Lower Score:**

N/A

---

### Decision · Program_Chairs · 2024-01-16

Reject